# Orthogonalizing Convolutional Layers with the Cayley Transform

**Asher Trockman**
Computer Science Department
Carnegie Mellon University
ashert@cs.cmu.edu

**J. Zico Kolter**
Computer Science Department
Carnegie Mellon University &
Bosch Center for AI
zkolter@cs.cmu.edu

## Abstract

Recent work has highlighted several advantages of enforcing *orthogonality* in the weight layers of deep networks, such as maintaining the stability of activations, preserving gradient norms, and enhancing adversarial robustness by enforcing low Lipschitz constants. Although numerous methods exist for enforcing the orthogonality of fully-connected layers, those for convolutional layers are more heuristic in nature, often focusing on penalty methods or limited classes of convolutions. In this work, we propose and evaluate an alternative approach to directly parameterize convolutional layers that are constrained to be orthogonal. Specifically, we propose to apply the Cayley transform to a skew-symmetric convolution in the Fourier domain, so that the inverse convolution needed by the Cayley transform can be computed efficiently. We compare our method to previous Lipschitz-constrained and orthogonal convolutional layers and show that it indeed preserves orthogonality to a high degree even for large convolutions. Applied to the problem of certified adversarial robustness, we show that networks incorporating the layer outperform existing deterministic methods for certified defense against $\ell_2$-norm-bounded adversaries, while scaling to larger architectures than previously investigated. Code is available at https://github.com/locuslab/orthogonal-convolutions.

## 1 Introduction

Encouraging *orthogonality* in neural networks has proven to yield several compelling benefits. For example, orthogonal initializations allow extremely deep vanilla convolutional neural networks to be trained quickly and stably (Xiao et al., 2018; Saxe et al., 2013). And initializations that remain closer to orthogonality throughout training seem to learn faster and generalize better (Pennington et al., 2017). Unlike Lipschitz-constrained layers, orthogonal layers are *gradient-norm-preserving* (Anil et al., 2019), discouraging vanishing and exploding gradients and stabilizing activations (Rodríguez et al., 2017). Orthogonality is thus a potential alternative to batch normalization in CNNs and can help to remember long-term dependencies in RNNs (Arjovsky et al., 2016; Vorontsov et al., 2017). Constraints and penalty terms encouraging orthogonality can improve generalization in practice (Bansal et al., 2018; Sedghi et al., 2018), improve adversarial robustness by enforcing low Lipschitz constants, and allow deterministic certificates of robustness (Tsuzuku et al., 2018).

Despite evidence for the benefits of orthogonality constraints, and while there are many methods to orthogonalize fully-connected layers, the orthogonalization of convolutions has posed challenges. More broadly, current Lipschitz-constrained convolutions rely on spectral normalization and kernel reshaping methods (Tsuzuku et al., 2018), which only allow loose bounds and can cause vanishing gradients. Sedghi et al. (2018) showed how to clip the singular values of convolutions and thus enforce orthogonality, but relied on costly alternating projections to achieve tight constraints. Most recently, Li et al. (2019) introduced the Block Convolution Orthogonal Parameterization (BCOP), which cannot express the full space of orthogonal convolutions.

In contrast to previous work, we provide a *direct*, expressive, and scalable parameterization of orthogonal convolutions. Our method relies on the Cayley transform, which is well-known for parameterizing orthogonal matrices in terms of skew-symmetric matrices, and can be easily extended to

non-square weight matrices. The transform requires efficiently computing the *inverse* of a particular convolution in the Fourier domain, which we show works well in practice.

We demonstrate that our Cayley layer is indeed orthogonal in practice when implemented in 32-bit precision, irrespective of the number of channels. Further, we compare it to alternative convolutional and Lipschitz-constrained layers: we include them in several architectures and evaluate their *deterministic* certifiable robustness against an $\ell_2$-norm-bounded adversary. Our layer provides state-of-the-art results on this task. We also demonstrate that the layers empirically endow a considerable degree of robustness without adversarial training. Our layer generally outperforms the alternatives, particularly for larger architectures.

## 2 Related Work

**Orthogonality in neural networks.** The benefits of orthogonal weight initializations for *dynamical isometry*, *i.e.*, ensuring signals propagate through deep networks, are explained by Saxe et al. (2013) and Pennington et al. (2017), with limited theoretical guarantees investigated by Hu et al. (2020). Xiao et al. (2018) provided a method to initialize orthogonal convolutions, and demonstrated that it allows the training of extremely deep CNNs without batch normalization or residual connections. Further, Qi et al. (2020) developed a novel regularization term to encourage orthogonality throughout training and showed its effectiveness for training very deep vanilla networks. The signal-preserving properties of orthogonality can also help with remembering long-term dependencies in RNNs, on which there has been much work (Helfrich et al., 2018; Arjovsky et al., 2016).

One way to orthogonalize weight matrices is with the *Cayley transform*, which is often used in Riemannian optimization (Absil et al., 2009). Helfrich et al. (2018) and Maduranga et al. (2019) avoid vanishing/exploding gradients in RNNs using the scaled Cayley transform. Similarly, Lezcano-Casado & Martínez-Rubio (2019) use the exponential map, which the Cayley transform approximates. Li et al. (2020) derive an iterative approximation of the Cayley transform for orthogonally-constrained optimizers and show it speeds the convergence of CNNs and RNNs. However, they merely orthogonalize a matrix obtained by reshaping the kernel, which is not the same as an orthogonal convolution (Sedghi et al., 2018). Our contribution is unique here in that we *parameterize orthogonal convolutions directly*, as opposed to reshaping kernels.

**Bounding neural network Lipschitzness.** Orthogonality imposes a strict constraint on the Lipschitz constant, which itself comes with many benefits: Lower Lipschitz constants are associated with improved robustness (Yang et al., 2020) and better generalization bounds (Bartlett et al., 2017). Tsuzuku et al. (2018) showed that neural network classifications can be *certified* as robust to $\ell_2$-norm-bounded perturbations given a Lipschitz bound and sufficiently confident classifications. Along with Szegedy et al. (2013), they noted that the Lipschitz constant of neural networks can be bounded if the constants of the layers are known. Thus, there is substantial work on Lipschitz-constrained and regularized layers, which we review in Sec. 5. However, Anil et al. (2019) realized that mere Lipschitz constraints can attenuate gradients, unlike orthogonal layers.

There have been other ideas for calculating and controlling the *minimal* Lipschitzness of neural networks, *e.g.*, through regularization (Hein & Andriushchenko, 2017), extreme value theory (Weng et al., 2018), or using semi-definite programming (Latorre et al., 2020; Chen et al., 2020; Fazlyab et al., 2019), but constructing bounds from Lipschitz-constrained layers is more scalable and efficient. Besides Tsuzuku et al. (2018)'s strategy for deterministic certifiable robustness, there are many approaches to deterministically verifying neural network defenses using SMT solvers (Huang et al., 2017; Ehlers, 2017; Carlini & Wagner, 2017), integer programming approaches (Lomuscio & Maganti, 2017; Tjeng & Tedrake, 2017; Cheng et al., 2017), or semi-definite programming (Raghunathan et al., 2018). Wong et al. (2018)'s approach to minimize an LP-based bound on the robust loss is more scalable, but networks made from Lipschitz-constrained components can be more efficient still, as shown by Li et al. (2019) who outperform their approach. However, none of these methods yet perform as well as probabilistic methods (Cohen et al., 2019).

Consequently, orthogonal layers appear to be an important component to enhance the convergence of deep networks while encouraging robustness and generalization.

## 3 BACKGROUND

**Orthogonality.** Since we are concerned with orthogonal convolutions, we review orthogonal matrices: A matrix $Q \in \mathbb{R}^{n \times n}$ is orthogonal if $Q^T Q = Q Q^T = I$. However, in building neural networks, layers do not always have equal input and output dimensions: more generally, a matrix $U \in \mathbb{R}^{m \times n}$ is *semi-orthogonal* if $U^T U = I$ or $U U^T = I$. Importantly, if $m \geq n$, then $U$ is also norm-preserving: $\|Ux\|_2 = \|x\|_2$ for all $x \in \mathbb{R}^n$. If $m < n$, then the mapping is merely non-expansive (a contraction), i.e., $\|Ux\|_2 \leq \|x\|_2$. A matrix having all singular values equal to 1 is orthogonal, and vice versa.

**Orthogonal convolutions.** The same concept of orthogonality applies to convolutional layers, which are also linear transformations. A convolutional layer $\mathsf{conv} : \mathbb{R}^{c \times n \times n} \to \mathbb{R}^{c \times n \times n}$ with $c = c_{\mathsf{in}} = c_{\mathsf{out}}$ input and output channels is orthogonal if and only if $\|\mathsf{conv}(X)\|_F = \|X\|_F$ for all input tensors $X \in \mathbb{R}^{c \times n \times n}$; the notion of semi-orthogonality extends similarly for $c_{\mathsf{in}} \neq c_{\mathsf{out}}$. Note that orthogonalizing each convolutional kernel as in Lezcano-Casado & Martínez-Rubio (2019); Lezcano-Casado (2019) does not yield an orthogonal (norm-preserving) convolution.

**Lipschitzness under the $\ell_2$ norm.** A consequence of orthogonality is 1-Lipschitzness. A function $f : \mathbb{R}^n \to \mathbb{R}^m$ is $L$-Lipschitz with respect to the $\ell_2$ norm iff $\|f(x) - f(y)\|_2 \leq L\|x - y\|_2$ for all $x, y \in \mathbb{R}^n$. If $L$ is the smallest such constant for $f$, then it's called the *Lipschitz constant* of $f$, denoted by $\mathsf{Lip}(f)$. An useful property for certifiable robustness is that the Lipschitz constant of the composition of $f$ and $g$ is upper-bounded by the product of their constants: $\mathsf{Lip}(f \circ g) \leq \mathsf{Lip}(f)\mathsf{Lip}(g)$. Since simple neural networks are fundamentally just composed functions, this allows us to bound their Lipschitz constants, albeit loosely. We can extend this idea to residual networks using the fact that $\mathsf{Lip}(f + g) \leq \mathsf{Lip}(f) + \mathsf{Lip}(g)$, which motivates using a convex combination in residual connections. More details can be found in Li et al. (2019); Szegedy et al. (2013).

**Lipschitz bounds for provable robustness.** If we know the Lipschitz constant of the neural network, we can certify that a classification with sufficiently a large margin is robust to $\ell_2$ perturbations below a certain magnitude. Specifically, denote the *margin* of a classification with label $t$ as

$$\mathcal{M}_f(x) = \max(0, y_t - \max_{i \neq t} y_i), \tag{1}$$

which can be interpreted as the distance between the correct logit and the next largest logit. Then if the logit function $f$ has Lipschitz constant $L$, and $\mathcal{M}_f(x) > \sqrt{2}L\epsilon$, then $f(x)$ is certifiably robust to perturbations $\{\delta : \|\delta\|_2 \leq \epsilon\}$. Tsuzuku et al. (2018) and Li et al. (2019) provide proofs.

## 4 THE CAYLEY TRANSFORM OF A CONVOLUTION

Before describing our method, we first review discrete convolutions and the Cayley transform; then, we show the need for inverse convolutions and how to compute them efficiently in the Fourier domain, which lets us parameterize orthogonal convolutions via the Cayley transform. The key idea in our method is that multi-channel convolution in the Fourier domain reduces to a batch of matrix-vector products, and making each of those matrices orthogonal makes the convolution orthogonal. We describe our method in more detail in Appendix A and provide a minimal implementation in PyTorch in Appendix E.

An unstrided convolutional layer with $c_{\mathsf{in}}$ input channels and $c_{\mathsf{out}}$ output channels has a weight tensor $W$ of shape $\mathbb{R}^{c_{\mathsf{out}} \times c_{\mathsf{in}} \times n \times n}$ and takes an input $X$ of shape $\mathbb{R}^{c_{\mathsf{in}} \times n \times n}$ to produce an output $Y$ of shape $\mathbb{R}^{c_{\mathsf{out}} \times n \times n}$, i.e., $\mathsf{conv}_W : \mathbb{R}^{c_{\mathsf{in}} \times n \times n} \to \mathbb{R}^{c_{\mathsf{out}} \times n \times n}$. It is easiest to analyze convolutions when they are *circular*: if the kernel goes out of bounds of $X$, it wraps around to the other side—this operation can be carried out efficiently in the Fourier domain. Consequently, we focus on circular convolutions.

We define $\mathsf{conv}_W(X)$ as the circular convolutional layer with weight tensor $W \in \mathbb{R}^{c_{\mathsf{out}} \times c_{\mathsf{in}} \times n \times n}$ applied to an input tensor $X \in \mathbb{R}^{c_{\mathsf{in}} \times n \times n}$ yielding an output tensor $Y = \mathsf{conv}_W(X) \in \mathbb{R}^{c_{\mathsf{out}} \times n \times n}$. Equivalently, we can view $\mathsf{conv}_W(X)$ as the doubly block-circulant matrix $C \in \mathbb{R}^{c_{\mathsf{out}} n^2 \times c_{\mathsf{in}} n^2}$ corresponding to the circular convolution with weight tensor $W$ applied to the unrolled input tensor $\mathsf{vec}\, X \in \mathbb{R}^{c_{\mathsf{in}} n^2 \times 1}$. Similarly, we denote by $\mathsf{conv}_W^T(X)$ the transpose $C^T$ of the same convolution, which can be obtained by transposing the first two channel dimensions of $W$ and flipping each of the last two (kernel) dimensions vertically and horizontally, calling the result $W'$, and computing $\mathsf{conv}_{W'}(X)$. We denote $\mathsf{conv}_W^{-1}(X)$ as the inverse of the convolution, i.e., with corresponding matrix $C^{-1}$, which is more difficult to efficiently compute.

Now we review how to perform a convolution in the spatial domain. We refer to a *pixel* as a $c_{\text{in}}$ or $c_{\text{out}}$-dimensional slice of a tensor, like $X[:, i, j]$. Each of the $n^2$ $(i, j)$ output pixels $Y[:, i, j]$ are computed as follows: for each $c \in [c_{\text{out}}]$, compute $Y[c, i, j]$ by centering the tensor $W[c]$ on the $(i, j)^{th}$ pixel of the input and taking a dot product, wrapping around pixels of $W$ that go out-of-bounds. Typically, $W$ is zero except for a $k \times k$ region of the last two (spatial) dimensions, which we call the *kernel* or the *receptive field*. Typically, convolutional layers have small kernels, *e.g.*, $k = 3$.

Considering now matrices instead of tensors, the *Cayley transform* is a bijection between skew-symmetric matrices $A$ and orthogonal matrices $Q$ without $-1$ eigenvalues:

$$Q = (I - A)(I + A)^{-1}. \tag{2}$$

A matrix is skew-symmetric if $A = -A^T$, and we can *skew-symmetrize* any square matrix $B$ by computing the skew-symmetric part $A = B - B^T$. The Cayley transform of such a skew-symmetric matrix is always orthogonal, which can be seen by multiplying $Q$ by its transpose and rearranging.

We can also apply the Cayley transform to convolutions, noting they are also linear transformations that can be represented as doubly block circulant matrices. While it is possible to construct the matrix $C$ corresponding to a convolution $\text{conv}_W$ and apply the Cayley transform to it, this is highly inefficient in practice: Convolutions can be easily skew-symmetrized by computing $\text{conv}_W(X) - \text{conv}_W^T(X)$, but finding their inverse is challenging; instead, we manipulate convolutions in the Fourier domain, taking advantage of the *convolution theorem* and the efficiency of the fast Fourier transform.

According to the *2D convolution theorem* (Jain, 1989), the circular convolution of two matrices in the Fourier domain is simply their elementwise product. We will show that the convolution theorem extends to multi-channel convolutions of tensors, in which case convolution reduces to a batch of complex matrix-vector products rather than elementwise products: inverting these smaller matrices is equivalent to inverting the convolution, and finding their skew-Hermitian part is equivalent to skew-symmetrizing the convolution, which allows us to compute the Cayley transform.

We define the 2D Discrete (Fast) Fourier Transform for tensors of order $\geq 2$ as a mapping $\text{FFT}$ : $\mathbb{R}^{m_1 \times \dots \times m_r \times n \times n} \to \mathbb{C}^{m_1 \times \dots \times m_r \times n \times n}$ defined by $\text{FFT}(X)[i_1, \dots, i_r] = F_n X[i_1, \dots, i_r] F_n$ for $i_l \in 1, \dots, m_l$ and $l \in 1, \dots, r$ and $r \geq 0$, where $F_n[i, j] = \frac{1}{\sqrt{n}} \exp(\frac{-2\pi i}{n})^{(i-1)(j-1)}$. That is, we treat all but the last two dimensions as batch dimensions. We denote $\tilde{X} = \text{FFT}(X)$ for a tensor $X$.

Using the convolution theorem, in the Fourier domain the $c^{th}$ output channel is the sum of the elementwise products of the $c_{\text{in}}$ input and weight channels: that is, $\tilde{Y}[c] = \sum_{k=1}^{c_{\text{in}}} \tilde{W}[c, k] \odot \tilde{X}[k]$. Equivalently, working in the Fourier domain, the $(i, j)^{th}$ pixel of the $c^{th}$ output channel is the dot product of the $(i, j)^{th}$ pixel of the $c^{th}$ weight with the $(i, j)^{th}$ input pixel: $\tilde{Y}[c, i, j] = \tilde{W}[c, :, i, j] \cdot \tilde{X}[:, i, j]$. From this, we can see that the whole $(i, j)^{th}$ Fourier-domain output pixel is the matrix-vector product

$$\text{FFT}(\text{conv}_W(X))[:, i, j] = \tilde{W}[:, :, i, j]\tilde{X}[:, i, j]. \tag{3}$$

This interpretation gives a way to compute the inverse convolution as required for the Cayley transform, assuming $c_{\text{in}} = c_{\text{out}}$:

$$\text{FFT}(\text{conv}_W^{-1}(X))[:, i, j] = \tilde{W}[:, :, i, j]^{-1}\tilde{X}[:, i, j]. \tag{4}$$

Given this method to compute inverse convolutions, we can now parameterize an orthogonal convolution with a skew-symmetric convolution through the Cayley transform, highlighted in Algorithm 1: In line 1, we use the Fast Fourier Transform on the weight and input tensors. In line 4, we compute the Fourier domain weights for the skew-symmetric convolution (the *Fourier representation* is skew-Hermitian, thus the use of the conjugate transpose). Next, in lines 4–5 we compute the inverses required for $\text{FFT}(\text{conv}_{I+A}^{-1}(x))$ and use them to compute the Cayley transform written as $(I+A)^{-1} - A(I+A)^{-1}$ in line 6. Finally, we get our spatial domain result with the inverse FFT, which is always *exactly* real despite working with complex matrices in the Fourier domain (see Appendix A).

### 4.1 Properties of our approach

It is important to note that *the inverse in the Cayley transform always exists*: Because $A$ is skew-symmetric, it has all imaginary eigenvalues, so $I + A$ has all nonzero eigenvalues and is thus nonsingular. Since only square matrices can be skew-symmetrized and inverted, Algorithm 1 only

---

**Algorithm 1:** Orthogonal convolution via the Cayley transform.

---

**Input:** A tensor $X \in \mathbb{R}^{c_{in} \times n \times n}$ and convolution weights $W \in \mathbb{R}^{c_{out} \times c_{in} \times n \times n}$, with $c_{in} = c_{out}$.
**Output:** A tensor $Y \in \mathbb{R}^{c_{out} \times n \times n}$, the orthogonal convolution parameterized by $W$ applied to $X$.

1   $\tilde{W} := \mathsf{FFT}(W) \in \mathbb{C}^{c_{out} \times c_{in} \times n \times n}, \tilde{X} := \mathsf{FFT}(X) \in \mathbb{C}^{c_{in} \times n \times n}$
2   **for** *all* $i, j \in 1, \ldots, n$ // In parallel
3   **do**
4      $\tilde{A}[:, :, i, j] := \tilde{W}[:, :, i, j] - \tilde{W}[:, :, i, j]^*$
5      $\tilde{Y}[:, i, j] := (I + \tilde{A}[:, :, i, j])^{-1} \tilde{X}[:, i, j]$
6      $\tilde{Z}[:, i, j] := \tilde{Y}[:, i, j] - \tilde{A}[:, :, i, j]\tilde{Y}[:, i, j]$
7   **end**
8   **return** $\mathsf{FFT}^{-1}(\tilde{Z}).real$

---

works for $c_{in} = c_{out}$, but can be extended to the rectangular case where $c_{out} \geq c_{in}$ by padding the matrix with zeros and then projecting out the first $c_{in}$ columns after the transform, resulting in a norm-preserving semi-orthogonal matrix; the case $c_{in} \geq c_{out}$ follows similarly, but the resulting matrix is merely non-expansive. With efficient implementation in terms of the Schur complement (Appendix A.1, Eq. A22), this only requires inverting a square matrix of order $\min(c_{in}, c_{out})$.

We saw that learning was easier if we parameterized $W$ in Algorithm 1 by $W = gV/\|V\|_F$ for a learnable scalar $g$ and tensor $V$, as in weight normalization (Salimans & Kingma, 2016).

**Comparison to BCOP.** While the Block Convolution Orthogonal Parameterization (BCOP) can only express orthogonal convolutions with fixed $k \times k$-sized kernels, a Cayley convolutional layer can represent orthogonal convolutions with a learnable kernel size up to the input size, and it does this without costly projections unlike Sedghi et al. (2018). However, our parameterization as presented is limited to orthogonal convolutions without -1 eigenvalues. Hence, our parameterization is *incomplete*; besides kernel size restrictions, BCOP was also demonstrated to incompletely represent the space of orthogonal convolutions, though the details of the problem were unresolved (Li et al., 2019).

Our method can represent such orthogonal convolutions by multiplying the Cayley transform by a fixed diagonal matrix with $\pm 1$ entries (Gallier, 2006; Helfrich et al., 2018); however, we cannot optimize over the discrete set of such scaling matrices, so our method cannot optimize over *all* orthogonal convolutions, nor all special orthogonal convolutions. In our experiments, we did not find improvements from adding randomly initialized scaling matrices as in Helfrich et al. (2018).

**Limitations of our method.** As our method requires computing an inverse convolution, it is generally incompatible with strided convolutions; *e.g.*, a convolution with stride 2 cannot be inverted since it involves noninvertible downsampling. It is possible to apply our method to stride-2 convolutions by simultaneously increasing the number of output channels by $4\times$ to compensate for the $2\times$ downsampling of the two spatial dimensions, though we found this to be computationally inefficient. Instead, we use the invertible downsampling layer from (Jacobsen et al., 2018) to emulate striding.

The convolution resulting from our method is *circular*, which is the same as using the circular padding mode instead of zero padding in, *e.g.*, PyTorch, and will not have a large impact on performance if subjects tend to be centered in images in the data set. BCOP (Li et al., 2019) and Sedghi et al. (2018) also restricted their attention to circular convolutions.

Our method is substantially more expensive than plain convolutional layers, though in most practical settings it is more efficient than existing work: We plot the runtimes of our Cayley layer, BCOP, and plain convolutions in a variety of settings in Figure 6 for comparison, and we also report runtimes in Tables 4 and 5 (see Appendix C).

**Runtime comparison** Our Cayley layer does $c_{in}c_{out}$ FFTs on $n \times n$ matrices (*i.e.*, the kernels padded to the input size), and $c_{in}$ FFTs for each $n \times n$ input. These have complexity $\mathcal{O}(c_{in}c_{out}n^2 \log n)$ and $\mathcal{O}(c_{out}n^2 \log n)$ respectively. The most expensive step is computing the inverse of $n^2$ square matrices of order $c = \min(c_{in}, c_{out})$, with complexity $\mathcal{O}(n^2c^3)$, similarly to the method of Sedghi et al. (2018). We note like the authors that parallelization could effectively make this $\mathcal{O}(n^2 \log n + c^3)$, and it is quite feasible in practice. As in Li et al. (2020), the inverse could be replaced with an iterative approximation, but we did not find it necessary for our relatively small architectures.

For comparison, the related layers BCOP and RKO (Sec. 5) take only $\mathcal{O}(c^3)$ to orthogonalize the convolution, and OSSN takes $\mathcal{O}(n^2 c^3)$ (Li et al., 2019). In practice, we found our Cayley layer takes anywhere from $1/2 \times$ to $4 \times$ as long as BCOP, depending on the architecture (see Appendix C).

## 5 EXPERIMENTS

Our experiments have two goals: First, we show that our layer remains orthogonal in practice. Second, we compare the performance of our layer versus alternatives (particularly BCOP) on two adversarial robustness tasks on CIFAR-10: We investigate the *certifiable* robustness against an $\ell_2$-norm-bounded adversary using the idea of Lipschitz Margin Training (Tsuzuku et al., 2018), and then we look at robustness in practice against a powerful adversary. We find that our layer is always orthogonal and performs relatively well in the robustness tasks. Separately, we show our layer improves on the Wasserstein distance estimation task from Li et al. (2019) in Appendix D.2.

For alternative layers, we adopt the naming scheme for previous work on Lipschitz-constrained convolutions from Li et al. (2019), and we compare directly against their implementations. We outline the methods below.

**RKO.** A convolution can be represented as a matrix-vector product, *e.g.*, using a doubly block-circulant matrix and the unrolled input. Alternatively, one could stack each $k \times k$ receptive field, and multiply by the $c_{\text{out}} \times k^2 c_{\text{in}}$ reshaped kernel matrix (Cisse et al., 2017). The spectral norm of this reshaped matrix is bounded by the convolution's true spectral norm (Tsuzuku et al., 2018). Consequently, *reshaped kernel methods* orthogonalize this reshaped matrix, upper-bounding the singular values of the convolution by 1. Cisse et al. (2017) created a penalty term based on this matrix; instead, like Li et al. (2019), we orthogonalize the reshaped matrix directly, called **reshaped kernel orthogonalization (RKO)**. They used an iterative algorithm for orthogonalization (Björck & Bowie, 1971); for comparison, we implement **RKO** using the Cayley transform instead of Björck orthogonalization, called **CRKO**.

**OSSN.** A prevalent idea to constrain the Lipschitz constants of convolutions is to approximate the maximum singular value and normalize it out: Miyato et al. (2018) used the power method on the matrix $W$ associated with the convolution, *i.e.*, $s_{i+1} := W^T W s_i$, and $\sigma_{max} \approx \|W s_n\| / \|s_n\|$. Gouk et al. (2018) improved upon this idea by applying the power method directly to convolutions, using the transposed convolution for $W^T$. However, this **one-sided** *spectral normalization* is quite restrictive; dividing out $\sigma_{max}$ can make other singular values vanishingly small.

**SVCM.** Sedghi et al. (2018) showed how to exactly compute the singular values of convolutional layers using the Fourier transform before the SVD, and proposed a **singular value clipping method**. However, the clipped convolution can have an arbitrarily large kernel size, so they resorted to alternating projections between orthogonal convolutions and $k \times k$-kernel convolutions, which can be expensive. Like Li et al. (2019), we found that $\approx 50$ projections are needed for orthogonalization.

**BCOP.** The Block Convolution Orthogonal Parameterization extends the orthogonal initialization method of Xiao et al. (2018). It differentiably parameterizes $k \times k$ orthogonal convolutions with an orthogonal matrix and $2(k-1)$ symmetric projection matrices. The method only parameterizes the subspace of orthogonal convolutions with $k \times k$-sized kernels, but is quite expressive empirically. Internally, orthogonalization is done with the method by Björck & Bowie (1971).

Note that BCOP and SVCM are the only other orthogonal convolutional layers, and SVCM only for a large number of projections. RKO, CRKO, and OSSN merely upper-bound the Lipschitz constant of the layer by 1.

### 5.1 TRAINING AND ARCHITECTURAL DETAILS

**Training details.** For all experiments, we used CIFAR-10 with standard augmentation, *i.e.*, random cropping and flipping. Inputs to the model are always in the range $[0, 1]$; we implement normalization as a layer for compatibility with AutoAttack. For each architecture/convolution pair, we tried learning rates in $\{10^{-5}, 10^{-4}, 10^{-3}, 10^{-2}, 10^{-1}\}$, choosing the one with the best test accuracy. Most often, 0.001 is appropriate. We found that a piecewise triangular learning rate, as used in top performers in the DAWNBench competition (Coleman et al., 2017), performed best. Adam (Kingma & Ba, 2014) showed a significant improvement over plain SGD, and we used it for all experiments.

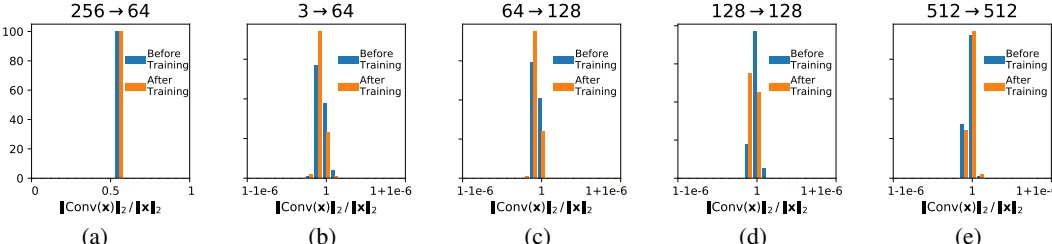

Figure 1: (Titles: $c_{\text{in}} \rightarrow c_{\text{out}}$). Our layer remains orthogonal in practice even for large convolutions (d, e), and is norm-preserving even when $c_{\text{out}} > c_{\text{in}}$ (b, c); it is nonexpansive when $c_{\text{in}} > c_{\text{out}}$ (a).

**Loss function.** Inspired by Tsuzuku et al. (2018), Anil et al. (2019) and Li et al. (2019) used multi-class hinge loss where the margin is the robustness certificate $\sqrt{2}L\epsilon_0$. We corroborate their finding that this works better than cross-entropy, and similarly use $\epsilon_0 = 0.5$. Varying $\epsilon_0$ controls a tradeoff between accuracy and robustness (see Fig. 5).

**Initialization.** We found that the standard uniform initialization in PyTorch performed well for our layer. We adjusted the variance, but significant differences required order-of-magnitude changes. For residual networks, we tried Fixup initialization (Zhang et al., 2019), but saw no significant improvement. We hypothesize this is due to (1) the learnable scaling parameter inside the Cayley transform, which changes significantly during training and (2) the dynamical isometry inherent with orthogonal layers. For alternative layers, we used the initializations from Li et al. (2019).

**Architecture considerations.** For fair comparison with previous work, we use the "large" network from Li et al. (2019), which was first implemented in Kolter & Wong (2017)'s work on certifiable robustness. We also compare the performance of the different layers in a 1-Lipschitz-constrained version of ResNet9 (He et al., 2016) and WideResNet10-10 (Zagoruyko & Komodakis, 2016). The architectures we could investigate were limited by compute and memory, as all the layers compared are relatively expensive. For RKO, OSSN, SVCM, and BCOP, we use Björck orthogonalization (Björck & Bowie, 1971) for fully-connected layers, as reported in Li et al. (2019); Anil et al. (2019). For our Cayley convolutional layer and CRKO, we orthogonalize the fully-connected layers with the Cayley transform to be consistent with our method. We found the gradient-norm-preserving GroupSort activation function from Anil et al. (2019) to be more effective than ReLU, and we used a group size of 2, *i.e.*, MaxMin.

**Strided convolutions.** For the KWLarge network, we used "invertible downsampling", which emulates striding by rearranging the inputs to have $4\times$ more channels while halving the two spatial dimensions and reducing the kernel size to $\lfloor k/2 \rfloor$ (Jacobsen et al., 2018). For the residual networks, we simply used a version of pooling, noting that average pooling is still non-expansive when multiplied by its kernel size, which allows us to use more of the network's capacity. We also halved the kernel size of the last pooling layer, instead adding another fully-connected layer; empirically, this resulted in higher local Lipschitz constants.

**Ensuring Lipschitz constraints.** Batch normalization layers scale their output, so they can't be included in our 1-Lipschitz-constrained architecture; the gradient-norm-preserving properties of our layers compensate for this. We ensure residual connections are non-expansive by making them a convex combination with a new learnable parameter $\alpha$, *i.e.*, $g(x) = \alpha f(x) + (1-\alpha)x$, for $\alpha \in [0, 1]$. To ensure the latter constraint, we use sigmoid($\alpha$). We can tune the overall Lipschitz bound to a given $L$ using the Lipschitz composition property, multiplying each of the $m$ layers by $L^{1/m}$.

## 5.2 Adversarial Robustness

For certifiable robustness, we report the fraction of certifiable test points: *i.e.*, those with classification margin $\mathcal{M}_f(\boldsymbol{x})$ greater than $\sqrt{2}L\epsilon$, where $\epsilon = 36/255$. For empirical defense, we use both vanilla projected gradient descent and AutoAttack by Croce & Hein (2020). For PGD, we use $\alpha = \epsilon/4.0$ with 10 iterations. Within AutoAttack, we use both APGD-CE and APGD-DLR, finding the decision-based attacks provided no improvements. We report on $\epsilon = 36/255$ for consistency with Li et al. (2019) and previous work on deterministic certifiable robustness (Wong et al., 2018). Additionally,

| | | **KWLarge**: Trained for *provable* robustness | | | | | | |
|---|---|---|---|---|---|---|---|---|
| $\epsilon$ | Test Acc. | Cayley | BCOP | RKO | CRKO | OSSN | SVCM | .85·Cayley |
| 0 | Clean | **75.33**±**.41** | 75.11±.37 | 74.47±.28 | 73.92±.27 | 71.69±.34 | 72.43±.84 | 74.35±.33 |
| $\frac{36}{255}$ | PGD | 67.66±.31 | 67.29±.35 | **68.32**±**.22** | 68.03±.28 | 65.13±.10 | 66.43±.62 | 67.29±.52 |
| | AutoAttack | 65.13±.48 | 64.62±.31 | **66.10**±**.26** | 65.95±.25 | 62.92±.16 | 64.27±.67 | 65.00±.58 |
| | *Certified* | **59.16**±**.36** | 58.29±.19 | 57.50±.17 | 57.48±.34 | 55.71±.57 | 52.11±.90 | 59.99±.40 |
| | Emp.Lip | 0.740±.01 | 0.740±.02 | 0.667±.01 | 0.668±.01 | 0.716±.01 | 0.570±.02 | 0.648±.01 |

Table 1: Trained without normalizing inputs, mean/s.d. from 5 experiments reported. Our Cayley layer outperforms other methods in both test and $\ell_2$ **certifiable robust accuracy.**

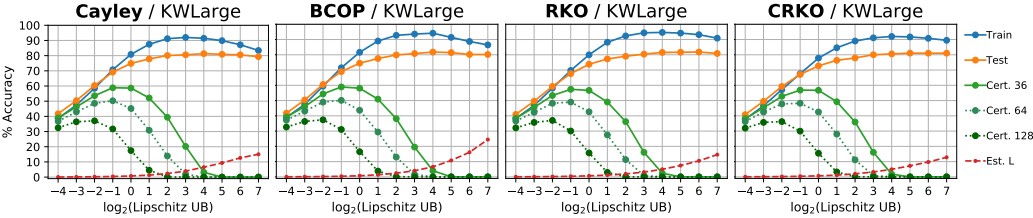

Figure 2: The **provable robustness** *vs.* clean accuracy tradeoff enabled by scaling the Lipschitz upper-bound for KWLarge.

we found it useful to report on empirical local Lipschitz constants throughout training using the PGD-like method from Yang et al. (2020).

### 5.3 RESULTS

**Practical orthogonality.** We show that our layer remains very close to orthogonality in practice, both before and after learning, when implemented in 32-bit precision. We investigated Cayley layers from one of our ResNet9 architectures, running them on random tensors to see if their norm is preserved, which is equivalent to orthogonality. We found that $\|\mathsf{Conv}(x)\|/\|x\|$, the extent to which our layer is gradient norm preserving, is always extremely close to 1. We illustrate the small discrepancies, easily bounded between 0.99999 and 1.00001, in Figure 1. Cayley layers which do not change or increase the number of channels are guaranteed to be orthogonal, which we see in practice for graphs (b, c, d, e). Those which decrease the number of channels can only be non-expansive, and in fact the layer seems to become slightly more norm-preserving after training (a). In short, our Cayley layer can capture the full benefits of orthogonality.

**Certifiable robustness.** We use our layer and alternatives within the KWLarge architecture for a more direct comparison to previous work on *deterministic* certifiable robustness (Li et al., 2019; Wong et al., 2018). As in (Li et al., 2019), we got the best performance without normalizing inputs, and can thus say that all networks compared here are at most 1-Lipschitz.

Our layer outperforms BCOP on this task (see Table 1), and is thus state-of-the-art, getting on average 75.33% clean test accuracy and 59.16% certifiable robust accuracy against adversarial perturbations with norm less than $\epsilon = 36/255$. In contrast, BCOP gets 75.11% test accuracy and 58.29% certifiable robust accuracy. The reshaped kernel methods perform only a percent or two worse on this task, while the spectral normalization and clipping methods lag behind.

We assumed that a layer is only meaningfully better than the other if both the test and robust accuracy are improved; otherwise, the methods may simply occupy different parts of the tradeoff curve. Since reshaped kernel methods can encourage smaller Lipschitz constants than orthogonal layers (Sedghi et al., 2018), we investigated the clean vs. certifiable robust accuracy tradeoff enabled by scaling the Lipschitz upper bound of the network, visualized in Figure 2. To that end, in light of the competitiveness of RKO, we chose a Lipschitz upper-bound of 0.85 which gave our Cayley layer similar test accuracy; this allowed for even higher certifiable robustness of 59.99%, but lower test

| $\epsilon$ | Test Acc. | ResNet9 | | | | WideResNet10-10 | | | |
|---|---|---|---|---|---|---|---|---|---|
| | | Cayley | BCOP | RKO | CRKO | Cayley | BCOP | RKO | CRKO |
| 0 | Clean | **81.70**±.12 | 80.72±.18 | 80.06±.15 | 79.38±.18 | **82.99** | 81.39 | 81.50 | 78.81 |
| $\frac{36}{255}$ | PGD | **73.77**±.19 | 73.27±.18 | 73.37±.12 | 72.52±.11 | **76.02** | 74.56 | 74.72 | 72.28 |
| | AutoAttack | **71.17**±.20 | 70.50±.06 | 71.01±.13 | 70.10±.07 | **73.16** | 71.86 | 72.24 | 69.97 |

Table 2: Empirical adversarial robustness for residual networks, mean and standard deviation for ResNet9 from 3 experiments. Cayley layers perform competitively on clean and robust accuracy.

accuracy of 74.35%. Overall, we were surprised by the similarity between the four top-performing methods after scaling Lipschitz constants.

We were not able to improve certifiable accuracy with ResNets. However, it was useful to increase the kernel size: we found 5 was an improvement in accuracy, while 7 and 9 were slightly worse. (Since our method operates in the Fourier domain, increases in kernel size incur no extra cost.) We also saw an improvement from scaling up the width of each layer of KWLarge, and our Cayley layer was substantially faster than BCOP as the width of KWLarge increased (see Appendix C). Multiplying the width by 3 and increasing the kernel size to 5, we were able to get 61.13% certified robust accuracy with our layer, and 60.55% with BCOP.

**Empirical robustness.** Previous work has shown that adversarial robustness correlates with lower Lipschitz constants. Thus, we investigated the robustness endowed by our layer against $\ell_2$ gradient-based adversaries. Here, we got better accuracy with the standard practice of normalizing inputs. Our layer outperformed the others in ResNet9 and WideResNet10-10 architectures; results were less decisive for KWLarge (see Appendix B). For the WideResNet, we got 82.99% clean accuracy and 73.16% robust accuracy for $\epsilon = 36/255$. For comparison, the state-of-the-art achieves 91.08% clean accuracy and 72.91% robust accuracy for $\epsilon = 0.5$ using a ResNet50 with adversarial training and additional unlabeled data (Augustin et al., 2020). We visualize the tradeoffs for our residual networks in Figure 3, noting that they empirically have smaller local Lipschitz constants than KWLarge. While our layer outperforms others for the default Lipschitz bound of 1, and is consistently slightly better than BCOP, RKO can perform similarly well for larger bounds. This provides some support for studies showing that hard constraints like ours may not match the performance of softer constraints, such as RKO and penalty terms (Bansal et al., 2018; Vorontsov et al., 2017).

## 6 CONCLUSION

In this paper, we presented a new, expressive parameterization of orthogonal convolutions using the Cayley transform. Unlike previous approaches to Lipschitz-constrained convolutions, ours gives deep networks the full benefits of orthogonality, such as gradient norm preservation. We showed empirically that our method indeed maintains a high degree of orthogonality both before and after learning, and also scales better to some architectures than previous approaches. Using our layer, we were able to improve upon the state-of-the-art in *deterministic* certifiable robustness against an $\ell_2$-norm-bounded adversary, and also showed that it endows networks with considerable inherent robustness empirically. While our layer offers benefits theoretically, we observed that heuristics involving orthogonalizing reshaped kernels were also quite effective for empirical robustness. Orthogonal convolutions may only show their true advantage in gradient norm preservation for deeper networks than we investigated. In light of our experiments in scaling the Lipschitz bound, we hypothesize that not orthogonality, but insead the ability of layers such as ours to exert control over the Lipschitz constant, may be best for the robustness/accuracy tradeoff. Future work may avoid expensive inverses using approximations or the exponential map, or compare various orthogonal and Lipschitz-constrained layers in the context of very deep networks.

ACKNOWLEDGMENTS

We thank Shaojie Bai, Chun Kai Ling, Eric Wong, and the anonymous reviewers for helpful feedback and discussions. This work was partially supported under DARPA grant number HR00112020006.

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

## A  Orthogonalizing Convolutions in the Fourier Domain

Our method relies on the fact that a multi-channel circular convolution can be *block-diagonalized* by a suitable Discrete Fourier Transform matrix. We show how this follows from the 2D convolution theorem (Jain, 1989, p. 145) below.

**Definition A.1.** $F_n$ is the DFT matrix for sequences of length $n$; we drop the subscript when it can be inferred from context.

**Definition A.2.** We define $\text{conv}_W(X)$ as in Section 4; if $c_{\text{in}} = c_{\text{out}} = 1$, we drop the channel axes, *i.e.*, for $X, W \in \mathbb{R}^{n \times n}$, the 2D circular convolution of $X$ with $W$ is $\text{conv}_W(X) \in \mathbb{R}^{n \times n}$.

**Theorem A.1.** *If $C \in \mathbb{R}^{n^2 \times n^2}$ represents a 2D circular convolution with weights $W \in \mathbb{R}^{n \times n}$ operating on a vectorized input $\text{vec}(X) \in \mathbb{R}^{n^2 \times 1}$, with $X \in \mathbb{R}^{n \times n}$, then it can be diagonalized as*

$$(F \otimes F)C(F^* \otimes F^*) = D.$$

*Proof.* According to the 2D convolution theorem, we can implement a single-channel 2D circular convolution by computing the elementwise product of the DFT of the filter and input signals:

$$FWF \odot FXF = F\text{conv}_W(X)F. \tag{A1}$$

This elementwise product is easier to work mathematically with if we represent it as a diagonal-matrix-vector product after vectorizing the equation:

$$\text{diag}(\text{vec}(FWF))\,\text{vec}(FXF) = \text{vec}(F\text{conv}_W(X)F). \tag{A2}$$

We can then rearrange this using $\text{vec}(ABC) = (C^T \otimes A)\,\text{vec}(B)$ and the symmetry of $F$:

$$\text{diag}(\text{vec}(FWF))(F \otimes F)\,\text{vec}(X) = (F \otimes F)\,\text{vec}(\text{conv}_W(X)). \tag{A3}$$

Left-multiplying by the inverse of $F \otimes F$ and noting $C\,\text{vec}(X) = \text{vec}(\text{conv}_W(X))$, we get the result

$$(F^* \otimes F^*)\,\text{diag}(\text{vec}(FWF))(F \otimes F) = C$$
$$\Rightarrow \qquad \text{diag}(\text{vec}(FWF)) = (F \otimes F)C(F^* \otimes F^*), \tag{A4}$$

which shows that the (doubly-block-ciculant) matrix $C$ is diagonalized by $F \otimes F$. An alternate proof can be found in Jain (1989, p. 150). $\qquad \square$

Now we can consider the case where we have a 2D circular convolution $\mathcal{C} \in \mathbb{R}^{c_{\text{out}}n^2 \times c_{\text{in}}n^2}$ with $c_{\text{in}}$ input channels and $c_{\text{out}}$ output channels. Here, $\mathcal{C}$ has $c_{\text{out}} \times c_{\text{in}}$ blocks, each of which is a circular convolution $\mathcal{C}_{ij} \in \mathbb{R}^{n^2 \times n^2}$. The input image is $\text{vec}\,\mathcal{X} = \left[\text{vec}^T X_1, \ldots, \text{vec}^T X_{c_{\text{in}}}\right]^T \in \mathbb{R}^{c_{\text{in}}n^2 \times 1}$, where $X_i$ is the $i^{th}$ channel of $\mathcal{X}$.

**Corollary A.1.1.** *If $\mathcal{C} \in \mathbb{R}^{c_{\text{out}}n^2 \times c_{\text{in}}n^2}$ represents a 2D circular convolution with $c_{\text{in}}$ input channels and $c_{\text{out}}$ output channels, then it can be block diagonalized as $\mathcal{F}_{c_{\text{out}}}\mathcal{C}\mathcal{F}_{c_{\text{in}}}^* = \mathcal{D}$, where $\mathcal{F}_c = S_{c,n^2}\left(I_c \otimes (F \otimes F)\right)$, $S_{c,n^2}$ is a permutation matrix, $I_k$ is the identity matrix of order $k$, and $\mathcal{D}$ is block diagonal with $n^2$ blocks of size $c_{\text{out}} \times c_{\text{in}}$.*

*Proof.* We first look at each of the blocks of $\mathcal{C}$ individually, referring to $\hat{\mathcal{D}}$ as the block matrix before applying the $S$ permutations, *i.e.*, $\hat{\mathcal{D}} = S_{c_{\text{out}},n^2}^T \mathcal{D} S_{c_{\text{in}},n^2}$, so that:

$$\hat{\mathcal{D}}_{ij} = [(I_{c_{\text{out}}} \otimes (F \otimes F))\,\mathcal{C}\,(I_{c_{\text{in}}} \otimes (F^* \otimes F^*))]_{ij} = (F \otimes F)\mathcal{C}_{ij}(F^* \otimes F^*)$$
$$= \text{diag}(\text{vec}(FW_{ij}F)), \tag{A5}$$

where $W_{ij}$ are the weights of the $(ij)^{\text{th}}$ single-channel convolution, using Theorem A.1. That is, $\hat{\mathcal{D}}$ is a block matrix of diagonal matrices. Then, let $S_{a,b}$ be the *perfect shuffle* matrix that permutes the block matrix of diagonal matrices to a block diagonal matrix. $S_{a,b}$ can be constructed by subselecting rows of the identity matrix. Using slice notation:

$$S_{a,b} = \begin{bmatrix} I_{ab}(1:b:ab,:) \\ I_{ab}(2:b:ab,:) \\ \vdots \\ I_{ab}(b:b:ab,:) \end{bmatrix}. \tag{A6}$$

As an example:

$$
S_{2,4}
\underbrace{\begin{bmatrix}
\mathbf{a}\ 0\ 0\ 0 & \mathbf{e}\ 0\ 0\ 0 & \mathbf{i}\ 0\ 0\ 0 \\
0\ \mathbf{b}\ 0\ 0 & 0\ \mathbf{f}\ 0\ 0 & 0\ \mathbf{j}\ 0\ 0 \\
0\ 0\ \mathbf{c}\ 0 & 0\ 0\ \mathbf{g}\ 0 & 0\ 0\ \mathbf{k}\ 0 \\
0\ 0\ 0\ \mathbf{d} & 0\ 0\ 0\ \mathbf{h} & 0\ 0\ 0\ \mathbf{l} \\
\mathbf{m}\ 0\ 0\ 0 & \mathbf{q}\ 0\ 0\ 0 & \mathbf{u}\ 0\ 0\ 0 \\
0\ \mathbf{n}\ 0\ 0 & 0\ \mathbf{r}\ 0\ 0 & 0\ \mathbf{v}\ 0\ 0 \\
0\ 0\ \mathbf{o}\ 0 & 0\ 0\ \mathbf{s}\ 0 & 0\ 0\ \mathbf{w}\ 0 \\
0\ 0\ 0\ \mathbf{p} & 0\ 0\ 0\ \mathbf{t} & 0\ 0\ 0\ \mathbf{x}
\end{bmatrix}}_{\hat{\mathcal{D}}}
S_{3,4}^{T}
=
\underbrace{\begin{bmatrix}
\mathbf{a}\ \mathbf{e}\ \mathbf{i} & 0\ 0\ 0 & 0\ 0\ 0 & 0\ 0\ 0 \\
\mathbf{m}\ \mathbf{q}\ \mathbf{u} & 0\ 0\ 0 & 0\ 0\ 0 & 0\ 0\ 0 \\
0\ 0\ 0 & \mathbf{b}\ \mathbf{f}\ \mathbf{j} & 0\ 0\ 0 & 0\ 0\ 0 \\
0\ 0\ 0 & \mathbf{n}\ \mathbf{r}\ \mathbf{v} & 0\ 0\ 0 & 0\ 0\ 0 \\
0\ 0\ 0 & 0\ 0\ 0 & \mathbf{c}\ \mathbf{g}\ \mathbf{k} & 0\ 0\ 0 \\
0\ 0\ 0 & 0\ 0\ 0 & \mathbf{o}\ \mathbf{s}\ \mathbf{w} & 0\ 0\ 0 \\
0\ 0\ 0 & 0\ 0\ 0 & 0\ 0\ 0 & \mathbf{d}\ \mathbf{h}\ \mathbf{l} \\
0\ 0\ 0 & 0\ 0\ 0 & 0\ 0\ 0 & \mathbf{p}\ \mathbf{t}\ \mathbf{x}
\end{bmatrix}}_{\mathcal{D}}.
\tag{A7}
$$

Then, with the perfect shuffle matrix, we can compute the block diagonal matrix $\mathcal{D}$ as:

$$
\begin{aligned}
S_{c_{\text{out}}, n^2} \hat{\mathcal{D}} S_{c_{\text{in}}, n^2}^{T} &= S_{c_{\text{out}}, n^2} \left( I_{c_{\text{out}}} \otimes (F \otimes F) \right) \mathcal{C} \left( I_{c_{\text{in}}} \otimes (F^* \otimes F^*) \right) S_{c_{\text{in}}, n^2}^{T} \\
&= \mathcal{F}_{c_{\text{out}}} \mathcal{C} \mathcal{F}_{c_{\text{in}}}^* = \mathcal{D}.
\end{aligned}
\tag{A8}
$$

The effect of left and right-multiplying with the perfect shuffle matrix is to create a new matrix $\mathcal{D}$ from $\hat{\mathcal{D}}$ such that $[\mathcal{D}_k]_{ij} = [\hat{\mathcal{D}}_{ij}]_{kk}$, where the subscript inside the brackets refers to the $k^{th}$ diagonal block and the $(ij)^{th}$ block respectively. $\qquad\square$

*Remark.* It is much more simple to compute $\mathcal{D}$ (here `wfft`) in tensor form given the convolution weights `w` as a $c_{\text{out}} \times c_{\text{in}} \times n \times n$ tensor:

```
wfft = fft2(w).reshape(cout, cin, n**2).permute(2, 0, 1).
```

**Definition A.3.** The Cayley transform is a bijection between skew-Hermitian matrices and unitary matrices; for real matrices, it is a bijection between skew-symmetric and orthogonal matrices. We apply the Cayley transform to an arbitrary matrix by first computing its skew-Hermitian part: we define the function $\mathsf{cayley} : \mathbb{C}^{m \times m} \to \mathbb{C}^{m \times m}$ by $\mathsf{cayley}(B) = (I_m - B + B^*)(I_m + B - B^*)^{-1}$, where we compute the skew-Hermitian part of $B$ inline as $B - B^*$. Note that the Cayley transform of a real matrix is always real, i.e., $\text{Im}(B) = 0 \Rightarrow \text{Im}(\mathsf{cayley}(B)) = 0$, in which case $B - B^* = B - B^T$ is a skew-symmetric matrix.

We now note a simple but important fact that we will use to show that our convolutions are always *exactly* real despite manipulating their complex representations in the Fourier domain.

**Lemma A.2.** *Say $J \in \mathbb{C}^{m \times m}$ is unitary so that $J^* J = I$, and $B = J \tilde{B} J^*$ for $B \in \mathbb{R}^{m \times m}$ and $\tilde{B} \in \mathbb{C}^{m \times m}$. Then $\mathsf{cayley}(B) = J\mathsf{cayley}(\tilde{B})J^*$.*

*Proof.* First note that $B = J\tilde{B}J^*$ implies $B^T = B^* = (J\tilde{B}J^*)^* = J\tilde{B}^*J^*$. Then

$$
\begin{aligned}
\mathsf{cayley}(B) = (I - B + B^T)(I + B - B^T)^{-1} &= (I - J\tilde{B}J^* + J\tilde{B}^*J^*)(I + J\tilde{B}J^* - J\tilde{B}^*J^*)^{-1} \\
&= J(I - \tilde{B} + \tilde{B}^*)J^* \left[ J(I + \tilde{B} - \tilde{B}^*)J^* \right]^{-1} \\
&= J(I - \tilde{B} + \tilde{B}^*)J^* \left[ J(I + \tilde{B} - \tilde{B}^*)^{-1}J^* \right] \\
&= J(I - \tilde{B} + \tilde{B}^*)(I + \tilde{B} - \tilde{B}^*)^{-1}J^* \\
&= J\mathsf{cayley}(\tilde{B})J^*.
\end{aligned}
\tag{A9}
$$
$\qquad\square$

For the rest of this section, we drop the subscripts of $\mathcal{F}$ and $S$ when they can be inferred from context.

**Theorem A.3.** *When $c_{\text{in}} = c_{\text{out}} = c$, applying the Cayley transform to the block diagonal matrix $\mathcal{D}$ results in a real, orthogonal multi-channel 2D circular convolution:*

$$
\mathsf{cayley}(\mathcal{C}) = \mathcal{F}^* \mathsf{cayley}(\mathcal{D}) \mathcal{F}.
$$

*Proof.* Note that $\mathcal{F}$ is unitary:

$$
\mathcal{F}\mathcal{F}^* = S(I_c \otimes (F \otimes F))(I_c \otimes (F^* \otimes F^*))S^T = SI_{cn^2}S^T = SS^T = I_{cn^2},
\tag{A10}
$$

since $S$ is a permutation matrix and is thus orthogonal. Then apply Lemma A.2, where we have $J = \mathcal{F}^*$, $B = \mathcal{C}$, and $\tilde{B} = \mathcal{D}$, to see the result. Note that $\mathsf{cayley}(\mathcal{C})$ is real because $\mathcal{C}$ is real; that is, even though we apply the Cayley transform to skew-Hermitian matrices in the Fourier domain, the resulting convolution is real. $\qquad\square$

*Remark.* While we deal with skew-Hermitian matrices in the Fourier domain, we are still effectively parameterizing the Cayley transform in terms of skew-symmetric matrices: as in the note in Lemma A.2, we can see that

$$\mathcal{C} = \mathcal{F}^*\mathcal{D}\mathcal{F} \Rightarrow \mathcal{C} - \mathcal{C}^{\mathcal{T}} = \mathcal{C} - \mathcal{C}^* = \mathcal{F}^*\mathcal{D}\mathcal{F} - \mathcal{F}^*\mathcal{D}^*\mathcal{F} = \mathcal{F}^*(D - D^*)\mathcal{F}, \tag{A11}$$

where $\mathcal{C}$ is real, $\mathcal{D}$ is complex, and $\mathcal{C} - \mathcal{C}^T$ is skew-symmetric (in the spatial domain) despite computing it with a skew-Hermitian matrix $\mathcal{D} - \mathcal{D}^*$ in the Fourier domain.

*Remark.* Since $\mathcal{D}$ is block diagonal, we only need to apply the Cayley transform (and thus invert) its $n^2$ blocks of size $c \times c$, which are much smaller than the whole matrix:

$$\mathsf{cayley}(\mathcal{D}) = \mathrm{diag}(\mathsf{cayley}(\mathcal{D}_1), \ldots, \mathsf{cayley}(\mathcal{D}_{n^2})). \tag{A12}$$

## A.1 Semi-Orthogonal Convolutions

In many cases, convolutional layers do not have $c_{\mathsf{in}} = c_{\mathsf{out}}$, in which case they cannot be orthogonal. Rather, we must resort to enforcing *semi-orthogonality*. We can semi-orthogonalize convolutions using the same techniques as above.

**Lemma A.4.** *Right-padding the multi-channel 2D circular convolution matrix $\mathcal{C}$ (from $c_{\mathsf{in}}$ to $c_{\mathsf{out}}$ channels) with $dn^2$ columns of zeros is equivalent to padding each diagonal block of the corresponding block-diagonal matrix $\mathcal{D}$ on the right with $d$ columns of zeros:*

$$[\mathcal{C} \quad \mathbf{0}_{dn^2}] = \mathcal{F}^* \, \mathrm{diag}([\mathcal{D}_1 \quad \mathbf{0}_d], \ldots, [\mathcal{D}_{n^2} \quad \mathbf{0}_d])\mathcal{F}, \tag{A13}$$

*where $\mathbf{0}_k$ refers to $k$ columns of zeros and a compatible number of rows.*

*Proof.* For a fixed column $j$, note that

$$[\mathcal{D}_k]_{ij} = 0 \text{ for all } i, k \iff [\hat{\mathcal{D}}_{ij}]_{kk} = 0 \text{ for all } i, k \iff \mathcal{C}_{ij} = \mathbf{0} \text{ for all } i, \tag{A14}$$

since $\hat{\mathcal{D}}_{ij} = (F \otimes F)\mathcal{C}_{ij}(F^* \otimes F^*) = \mathbf{0}$ only when $\mathcal{C}_{ij} = \mathbf{0}$. Apply this for $j = c_{\mathsf{in}}+1, \ldots, c_{\mathsf{in}}+d$. $\quad\square$

**Lemma A.5.** *Projecting out $d$ blocks of columns of $\mathcal{C}$ is equivalent to projecting out $d$ columns of each of the diagonal blocks of $\mathcal{D}$:*

$$\mathcal{C} \begin{bmatrix} I_{dn^2} \\ \mathbf{0} \end{bmatrix} = \mathcal{F}^* \, \mathrm{diag}\left(\mathcal{D}_1 \begin{bmatrix} I_d \\ \mathbf{0} \end{bmatrix}, \ldots, \mathcal{D}_{n^2} \begin{bmatrix} I_d \\ \mathbf{0} \end{bmatrix}\right) \mathcal{F} \tag{A15}$$

*Proof.* This proceeds similarly to the previous lemma: removing columns of each of the $n^2$ matrices $\mathcal{D}_1, \ldots, \mathcal{D}_{n^2}$ implies removing the corresponding blocks of columns of $\hat{\mathcal{D}}$, and thus of $\mathcal{C}$. $\quad\square$

**Theorem A.6.** *If $\mathcal{C}$ is a 2D multi-channel convolution with $c_{\mathsf{in}} \leq c_{\mathsf{out}}$, then letting $d = c_{\mathsf{out}} - c_{\mathsf{in}}$,*

$$\mathsf{cayley}\left([\mathcal{C} \quad \mathbf{0}_{dn^2}]\right) \begin{bmatrix} I_{dn^2} \\ \mathbf{0} \end{bmatrix} =$$

$$\mathcal{F}^* \, \mathrm{diag}\left(\mathsf{cayley}\left([\mathcal{D}_1 \quad \mathbf{0}_d]\right) \begin{bmatrix} I_d \\ \mathbf{0}_d \end{bmatrix}, \ldots, \mathsf{cayley}\left([\mathcal{D}_{n^2} \quad \mathbf{0}_d]\right) \begin{bmatrix} I_d \\ \mathbf{0}_d \end{bmatrix}\right) \mathcal{F}, \tag{A16}$$

*which is a real 2D multi-channel semi-orthogonal circular convolution.*

*Proof.* For the first step, we use Lemma A.4 for right padding, getting

$$[\mathcal{C} \quad \mathbf{0}_{dn^2}] = \mathcal{F}^* \, \mathrm{diag}([\mathcal{D}_1 \quad \mathbf{0}_d], \ldots, [\mathcal{D}_{n^2} \quad \mathbf{0}_d])\mathcal{F}. \tag{A17}$$

Then, noting that $[\mathcal{C} \quad \mathbf{0}_{dn^2}]$ is a convolution matrix with $c_{\mathsf{in}} = c_{\mathsf{out}}$, we can apply Theorem A.3 (and the following remark) to get:

$$\mathsf{cayley}\left([\mathcal{C} \quad \mathbf{0}_{dn^2}]\right) = \mathcal{F}^* \, \mathrm{diag}\left(\mathsf{cayley}\left([\mathcal{D}_1 \quad \mathbf{0}_d]\right), \ldots, \mathsf{cayley}\left([\mathcal{D}_{n^2} \quad \mathbf{0}_d]\right)\right) \mathcal{F}. \tag{A18}$$

Since $\mathsf{cayley}\left([\mathcal{C} \quad \mathbf{0}_{dn^2}]\right)$ is still a real convolution matrix, we can apply Lemma A.5 to get the result. $\quad\square$

This demonstrates that we can semi-orthogonalize convolutions with $c_{\text{in}} \neq c_{\text{out}}$ by first padding them so that $c_{\text{in}} = c_{\text{out}}$; despite performing padding, the Cayley transform, and projections on complex matrices in the Fourier domain, we have shown that the resulting convolution is still real. In practice, we do not literally perform padding nor projections; we explain how to do an equivalent but more efficient comptutation on each diagonal block $\mathcal{D}_k \in \mathbb{C}^{c_{\text{out}} \times c_{\text{in}}}$ below.

**Proposition A.7.** *We can efficiently compute the Cayley transform for semi-orthogonalization,* i.e., $\mathsf{cayley}\left(\begin{bmatrix} W & \mathbf{0}_d \end{bmatrix}\right) \begin{bmatrix} I_d \\ \mathbf{0}_d \end{bmatrix}$, *when* $c_{\text{in}} \leq c_{\text{out}}$ *by writing the inverse in terms of the Schur complement.*

*Proof.* We can partition $W \in \mathbb{C}^{c_{\text{out}} \times c_{\text{in}}}$ into its top part $U \in \mathbb{C}^{c_{\text{in}} \times c_{\text{in}}}$ and bottom part $V \in \mathbb{C}^{(c_{\text{out}} - c_{\text{in}}) \times c_{\text{in}}}$, and then write the padded matrix $\begin{bmatrix} W & \mathbf{0}_{c_{\text{out}} - c_{\text{in}}} \end{bmatrix} \in \mathbb{C}^{c_{\text{out}} \times c_{\text{out}}}$ as

$$\begin{bmatrix} W & \mathbf{0}_{c_{\text{out}} - c_{\text{in}}} \end{bmatrix} = \begin{bmatrix} U & \mathbf{0} \\ V & \mathbf{0} \end{bmatrix}. \tag{A19}$$

Taking the skew-Hermitian part and applying the Cayley transform, then projecting, we get:

$$\mathsf{cayley}\left(\begin{bmatrix} U & \mathbf{0} \\ V & \mathbf{0} \end{bmatrix}\right) \begin{bmatrix} I_{c_{\text{in}}} \\ \mathbf{0} \end{bmatrix} = \left(I_{c_{\text{out}}} - \begin{bmatrix} U & \mathbf{0} \\ V & \mathbf{0} \end{bmatrix} + \begin{bmatrix} U & \mathbf{0} \\ V & \mathbf{0} \end{bmatrix}^*\right) \left(I_{c_{\text{out}}} + \begin{bmatrix} U & \mathbf{0} \\ V & \mathbf{0} \end{bmatrix} - \begin{bmatrix} U & \mathbf{0} \\ V & \mathbf{0} \end{bmatrix}^*\right)^{-1} \begin{bmatrix} I_{c_{\text{in}}} \\ \mathbf{0} \end{bmatrix}$$

$$= \begin{bmatrix} I_{c_{\text{in}}} - U + U^* & V^* \\ -V & I_{c_{\text{out}} - c_{\text{in}}} \end{bmatrix} \begin{bmatrix} I_{c_{\text{in}}} + U - U^* & -V^* \\ V & I_{c_{\text{out}} - c_{\text{in}}} \end{bmatrix}^{-1} \begin{bmatrix} I_{c_{\text{in}}} \\ \mathbf{0} \end{bmatrix}. \tag{A20}$$

We focus on computing the inverse while keeping only the first $c_{\text{in}}$ columns. We use the inversion formula noted in Zhang (2006, p. 13) for a block partitioned matrix $M$,

$$M^{-1} \begin{bmatrix} I_{c_{\text{in}}} \\ \mathbf{0} \end{bmatrix} = \begin{bmatrix} P & Q \\ R & S \end{bmatrix}^{-1} \begin{bmatrix} I_{c_{\text{in}}} \\ \mathbf{0} \end{bmatrix}$$

$$= \begin{bmatrix} (M/S)^{-1} & -(M/S)^{-1} Q S^{-1} \\ -S^{-1} R (M/S)^{-1} & S^{-1} + S^{-1} R (M/S)^{-1} Q S^{-1} \end{bmatrix} \begin{bmatrix} I_{c_{\text{in}}} \\ \mathbf{0} \end{bmatrix}$$

$$= \begin{bmatrix} (M/S)^{-1} \\ -S^{-1} R (M/S)^{-1} \end{bmatrix}, \tag{A21}$$

where we assume $M$ takes the form of the inverse in Eq. A20, and $M/S = P - Q S^{-1} R$ is the Schur complement. Using this formula for the first $c_{\text{in}}$ columns of the inverse in Eq. A20, and computing the Schur complement $I_{c_{\text{in}}} + U - U^* + V^* I_{c_{\text{out}} - c_{\text{in}}}^{-1} V$, we find

$$\mathsf{cayley}\left(\begin{bmatrix} U & \mathbf{0} \\ V & \mathbf{0} \end{bmatrix}\right) = \begin{bmatrix} I_{c_{\text{in}}} - U + U^* & V^* \\ -V & I_{c_{\text{out}} - c_{\text{in}}} \end{bmatrix} \begin{bmatrix} (I_{c_{\text{in}}} + U - U^* + V^* V)^{-1} \\ -V (I_{c_{\text{in}}} + U - U^* + V^* V)^{-1} \end{bmatrix}$$

$$= \begin{bmatrix} (I_{c_{\text{in}}} - U + U^* - V^* V)(I_{c_{\text{in}}} + U - U^* + V^* V)^{-1} \\ -2V (I_{c_{\text{in}}} + U - U^* + V^* V)^{-1} \end{bmatrix} \in \mathbb{C}^{c_{\text{out}} \times c_{\text{in}}}, \tag{A22}$$

which is semi-orthogonal and requires computing only one inverse of size $c_{\text{in}} \leq c_{\text{out}}$. Note that this inverse always exists because $U - U^*$ is skew-Hermitian, so it has purely imaginary eigenvalues, and $V^* V$ is positive semidefinite and has all real non-negative eigenvalues. That is, the sum $I_{c_{\text{in}}} + U - U^* + V^* V$ has all nonzero eigenvalues and is thus nonsingular. $\qquad \square$

**Proposition A.8.** *We can also compute semi-orthogonal convolutions when* $c_{\text{in}} \geq c_{\text{out}}$ *using the method described above because* $\mathsf{cayley}\left(\begin{bmatrix} \mathcal{C}^T & \mathbf{0} \end{bmatrix}\right)^T = \mathsf{cayley}\left(\begin{bmatrix} \mathcal{C} \\ \mathbf{0} \end{bmatrix}\right)$.

*Proof.* We use that $(A^{-1})^T = (A^T)^{-1}$ and $(I - A)(I + A)^{-1} = (I + A)^{-1}(I - A)$ to see

$$\mathsf{cayley}\left(\begin{bmatrix} \mathcal{C} \\ \mathbf{0} \end{bmatrix}\right)^T = \left[\left(I - \begin{bmatrix} \mathcal{C} \\ \mathbf{0} \end{bmatrix} + \begin{bmatrix} \mathcal{C} \\ \mathbf{0} \end{bmatrix}^T\right) \left(I + \begin{bmatrix} \mathcal{C} \\ \mathbf{0} \end{bmatrix} - \begin{bmatrix} \mathcal{C} \\ \mathbf{0} \end{bmatrix}^T\right)^{-1}\right]^T$$

$$= \left(I + \begin{bmatrix} \mathcal{C} \\ \mathbf{0} \end{bmatrix}^T - \begin{bmatrix} \mathcal{C} \\ \mathbf{0} \end{bmatrix}\right)^{-1} \left(I - \begin{bmatrix} \mathcal{C} \\ \mathbf{0} \end{bmatrix}^T + \begin{bmatrix} \mathcal{C} \\ \mathbf{0} \end{bmatrix}\right)$$

$$= \mathsf{cayley}\left(\begin{bmatrix} \mathcal{C} \\ \mathbf{0} \end{bmatrix}^T\right) = \mathsf{cayley}\left(\begin{bmatrix} \mathcal{C}^T & \mathbf{0} \end{bmatrix}\right). \tag{A23}$$

$$\square$$

We have thus shown how to (semi-)orthogonalize real multi-channel 2D circular convolutions efficiently in the Fourier domain. A minimal implementation of our method can be found in Appendix E. The techniques described above could also be used with other orthogonalization methods, or for calculating the determinants or singular values of convolutions.

# B  ADDITIONAL RESULTS

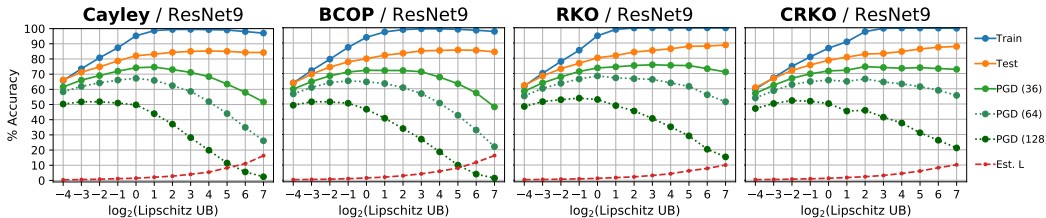

Figure 3: The robustness/accuracy tradeoff from changing the Lipschitz upper bound for ResNet9.

| | | **KWLarge**: Trained for test & PGD accuracy | | | | | | |
|---|---|---|---|---|---|---|---|---|
| $\epsilon$ | Test Acc. | Cayley | BCOP | RKO | CRKO | OSSN | SVCM | $.85\cdot$Cayley |
| 0 | Clean | 79.87±.33 | **80.14±.32** | 79.65±.20 | 79.53±.22 | 78.66±.21 | 79.19±.30 | 79.55±.19 |
| $\frac{36}{255}$ | PGD | 66.09±.47 | 65.31±.55 | **68.66±.20** | 68.55±.22 | 68.08±.20 | 68.35±.26 | 66.75±.34 |
| | AutoAttack | 62.02±.60 | 60.63±.72 | 65.16±.14 | **65.40±.28** | 64.90±.17 | 64.92±.15 | 62.67±.32 |
| | *Certified* | **38.67±.23** | 36.92±.30 | 34.67±.32 | 35.87±.43 | 36.29±.65 | 29.92±1.1 | 41.97±.23 |
| | Emp.Lip | 2.245±.04 | 2.272±.03 | 2.891±.02 | 1.866±.04 | 1.923±.06 | 1.698±.12 | 1.999±.03 |

Table 3: Trained with normalizing inputs, mean and standard deviation from 5 experiments reported. The normalization layer increases the Lipschitz bound of the network to $\approx 4.1$, *viz.*, CIFAR-10 standard deviation.

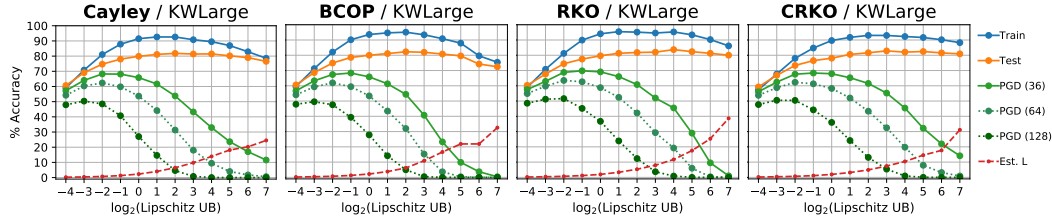

Figure 4: Robustness vs. accuracy tradeoff from changing the Lipschitz upper bound for KWLarge.

For KWLarge, our results on empirical robustness were mixed: while our Cayley layer outperforms BCOP in robust accuracy, the RKO methods are overall more robust by around 2%, for only a marginal decrease in clean accuracy. We note the lower empirical local Lipschitzness of RKO methods, which may explain their higher robustness: Figure 4 shows that the best choice of Lipschitz upper-bound for Cayley and BCOP layers may be less than 1 for this architecture.

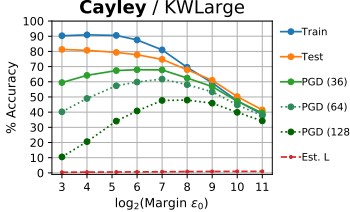

Figure 5: Effect of varying $\epsilon_0$ for Lipschitz margin training for KWLarge.

## C  EMPIRICAL RUNTIMES

| ↓ Layer | Width Mult. → | **KWLarge** Empirical Runtimes | | | | |
| --- | --- | --- | --- | --- | --- | --- |
| | | 1 | 2 | 3 | 6 | 8 |
| Cayley | Test Acc. | 75.97 | 76.97 | 76.94 | 76.80 | **78.28** |
| | Certified | 59.51 | 60.86 | **61.13** | 60.37 | 61.03 |
| | Avg. sec/Epoch | 9.01 | 17.50 | 32.30 | 133.43 | 260.31 |
| BCOP | Test Acc. | 75.28 | 75.81 | 76.35 | 76.55 | –.– |
| | Certified | 58.63 | 59.34 | 59.69 | 59.45 | –.– |
| | Avg. sec/Epoch | 20.75 | 53.72 | 135.95 | 1050.61 | –.– |
| RKO | Test Acc. | 74.85 | 75.74 | 76.05 | 76.29 | –.– |
| | Certified | 57.59 | 58.74 | 59.07 | 58.69 | –.– |
| | Avg. sec/Epoch | 16.02 | 50.15 | 131.43 | 1034.96 | –.– |

Table 4: Here we multiplied the input channels and output channels of each layer of KWLarge by $width$; we report on changes in accuracy and average runtime per epoch (100 epochs). Width 1 was on a Nvidia RTX 2080 Ti, while 2, 3, 6, and 8 were on a Nvidia Quadro RTX 8000. In this case, we were unable to scale BCOP to width 8 due to time and memory constraints. Generally, the wider networks may need more epochs to converge.

| Architecture | Avg. sec/Epoch | | | | |
| --- | --- | --- | --- | --- | --- |
| | Cayley | BCOP | RKO | Plain Conv. | Both Plain |
| ResNet9 | 43.92 | 48.04 | 45.73 | 14.56 | 13.27 |
| WideResNet10-10 | 210.6 | 109.4 | 99.46 | 48.40 | 46.10 |

Table 5: Our Cayley layer was not as fast for residual networks, possibly because they have convolutions with more channels and also larger spatial dimension, which is a multiplicative factor in our runtime analysis. This is especially true for the WideResNet. For *plain conv*, we replaced the Cayley convolutional layer with a plain circular convolution, leaving the Cayley fully-connected layers. For *both plain*, we also used plain fully-connected layers.

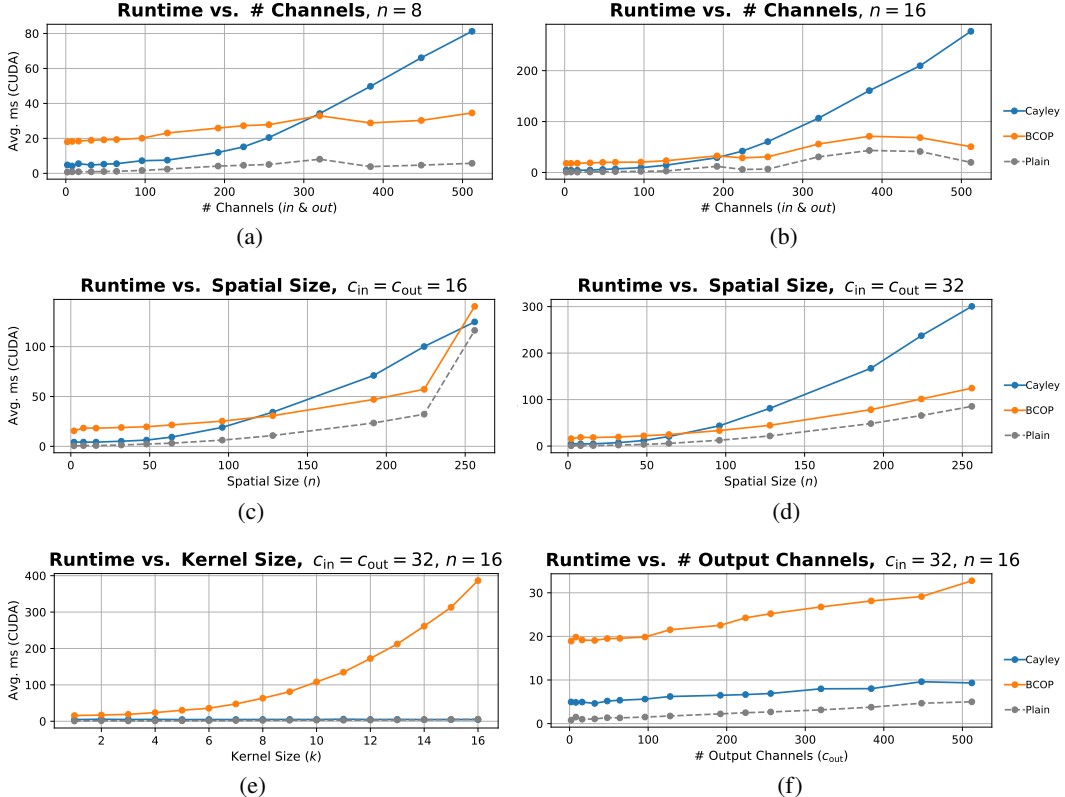

Figure 6: Our Cayley layer is particulaly efficient for inputs with small spatial dimension (width and height, *i.e.*, $n$) (see *(a), (c)*), large kernel size $k$ (see *(e)*), and cases where the number of input and output channels are not equal ($c_\text{in} \neq c_\text{out}$) (see *(f)*). For very large spatial size (image width and height) (see *(d)*), or the combination of relatively large spatial size and many channels (see *(b)*), BCOP (Li et al., 2019) tends to be more efficient. Since convolutional layers in neural networks tend to decrease the spatial dimensionality while increasing the number of channels, and also often have unequal numbers of input and output channels, our Cayley layer is often more efficient in practice. In all cases, orthogonal convolutional layers are significantly slower than plain convolutional layers.

Each runtime was recorded using the autograd profiler in PyTorch (Paszke et al., 2019) by summing the CUDA execution times. The batch size was fixed at 128 for all graphs, and each data point was averaged over 32 iterations. We used a Nvidia Quadro RTX 8000.

# D    ADDITIONAL BASELINE EXPERIMENTS

## D.1    ROBUSTNESS EXPERIMENTS

| | **KWLarge** | |
|---|---|---|
| $\epsilon$ | Test Acc. | CayleyBCOP |
| 0 | Clean | 74.35±.29 |
| $\frac{36}{255}$ | PGD | 66.91±.10 |
| | AutoAttack | 64.36±.17 |
| | *Certified* | 57.94±.22 |
| | Emp.Lip | 0.732±.01 |

Table 6: Additional baseline for KWLarge trained for provable robustness. Mean and s.d. over 5 trials.

| | **ResNet9** | | |
|---|---|---|---|
| $\epsilon$ | Test Acc. | Plain Conv. | CayleyBCOP |
| 0 | Clean | 89.22±.18 | 80.05±.20 |
| $\frac{36}{255}$ | PGD | 70.86±.06 | 72.58±.25 |
| | AutoAttack | 68.20±.10 | 69.90±.35 |

Table 7: Additional baselines for ResNet9 trained for empirical adversarial robustness. Mean and s.d. over 3 trials.

The main competing orthogonal convolutional layer, BCOP (Li et al., 2019), uses Björck (Björck & Bowie, 1971) orthogonalization for internal parameter matrices; they also used it in their experiments for orthogonal fully-connected layers. Similarly to how we replaced the  method in RKO with the Cayley transform for our CRKO (Cayley RKO) experiments, we replaced Björck with the Cayley transform in BCOP and used a Cayley linear layer for *CayleyBCOP* experiments, reported in Tables 6 and 7. We see slightly decreased performance over all metrics, similarly to the relationship between RKO and CRKO.

For additional comparison, we also report on a plain convolutional baseline in Table 7. For this model, we used a plain circular convolutional layer and a Cayley linear layer, which still imparts a considerable degree of robustness. With the plain convolutional layer, the model gains a considerable degree of accuracy but loses some robustness. We did not report a plain convolutional baseline for the provable robustness experiments on KWLarge, as it would require a more sophisticated technique to bound the Lipschitz constants of each layer, which is outside the scope of our investigation.

## D.2    WASSERSTEIN DISTANCE ESTIMATION

| | Cayley | BCOP | RKO | OSSN |
|---|---|---|---|---|
| **Wasserstein Distance:** | **10.72** | 10.08 | 9.18 | 7.50 |

Table 8: For BCOP, RKO and OSSN, we report the *best* bound over all trials from the experiments in the repository containing BCOP's implementation (Li et al., 2019). We ran *one trial* of the Wasserstein GAN experiment, replacing the BCOP and Björck layers with our Cayley convolutional and linear layers, and achieved a significantly tighter bound. We only report on experiments using the GroupSort (MaxMin) activation (Anil et al., 2019) and on the STL-10 dataset.

We repeated the Wasserstein distance estimation experiment from Li et al. (2019), simply replacing the BCOP layer with our Cayley convolutional layer, and the Björck linear layer with our Cayley fully-connected layer. We took the best Wasserstein distance bound from one trial of each of the four learning rates considered in BCOP (0.1, 0.01, 0.001, 0.0001); see Table 8.

## E    EXAMPLE IMPLEMENTATIONS

In PyTorch 1.8, our layer can be implemented as follows.

```python
def cayley(W):
    if len(W.shape) == 2:
        return cayley(W[None])[0]
    _, cout, cin = W.shape
    if cin > cout:
        return cayley(W.transpose(1, 2)).transpose(1, 2)
    U, V = W[:, :cin], W[:, cin:]
    I = torch.eye(cin, dtype=W.dtype, device=W.device)[None, :, :]
    A = U - U.conj().transpose(1, 2) + V.conj().transpose(1, 2) @ V
    inv = torch.inverse(I + A)
    return torch.cat((inv @ (I - A), -2 * V @ inv), axis=1)

class CayleyConv(nn.Conv2d):
    def fft_shift_matrix(self, n, s):
        shift = torch.arange(0, n).repeat((n, 1))
        shift = shift + shift.T
        return torch.exp(2j * np.pi * s * shift / n)

    def forward(self, x):
        cout, cin, _, _ = self.weight.shape
        batches, _, n, _ = x.shape
        if not hasattr(self, "shift_matrix"):
            s = (self.weight.shape[2] - 1) // 2
            self.shift_matrix = self.fft_shift_matrix(n, -s)[:, :(n//2 + 1)] \
                                    .reshape(n * (n // 2 + 1), 1, 1).to(x.device)
        xfft = torch.fft.rfft2(x).permute(2, 3, 1, 0) \
                .reshape(n * (n // 2 + 1), cin, batches)
        wfft = self.shift_matrix * torch.fft.rfft2(self.weight, (n, n)) \
                .reshape(cout, cin, n * (n // 2 + 1)).permute(2, 0, 1).conj()
        yfft = (cayley(wfft) @ xfft).reshape(n, n // 2 + 1, cout, batches)
        y = torch.fft.irfft2(yfft.permute(3, 2, 0, 1))
        if self.bias is not None:
            y += self.bias[:, None, None]
        return y
```

To make the layer support stride-2 convolutions, have `CayleyConv` inherit from the following class instead, which depends on the `einops` package:

```python
class StridedConv(nn.Conv2d):
    def __init__(self, *args, **kwargs):
        if "stride" in kwargs and kwargs["stride"] == 2:
            args = list(args)
            args[0] = 4 * args[0] # 4x in_channels
            args[2] = args[2] // 2 # //2 kernel_size; optional
            args = tuple(args)
        super().__init__(*args, **kwargs)
        downsample = "b c (w k1) (h k2) -> b (c k1 k2) w h"
        self.register_forward_pre_hook(lambda _, x: \
                einops.rearrange(x[0], downsample, k1=2, k2=2) \
                if self.stride == (2, 2) else x[0])
```

More details on our implementation and experiments can be found at:
https://github.com/locuslab/orthogonal-convolutions.

