# OpenReview forum: "Orthogonalizing Convolutional Layers with the Cayley Transform"
_ICLR.cc/2021/Conference — ICLR 2021 Spotlight_

### Official Review · AnonReviewer2 · 2020-10-14
**Short and nice idea. Very strong paper overall.**

**Rating:** 8
**Confidence:** 4

**Review:**

***Summary***
The paper uses the idea that a convolution on the Fourier space accounts just to a matrix-vector product. It then uses the Cayley map to parametrise the special orthogonal group, compute the action and send the tensor back using the inverse FFT. They test this idea in the context of certifiable robustness.

***Comments***

1) Page 3. "Computing a dot product".
Usually the term "dot product" is used to refer to scalar products. It would be perhaps more reasonable to say that it performs a "matrix-vector product".

2) Page 3. "Typically, $W$ is zero except for a $k\times k$ region, which we call the kernel or the receptive field.".
Here $W$ is the matrix version of the convolution. For this reason, $W$ is non zero in $k^2$ elements in every row of each matrix $W[:, :, i, j]$, not in a $k \times k$ region. Although I might have misunderstood this...

3) The operator $\operatorname{conv}$ is not defined anywhere. In general, it would be very helpful to say where does each object live in page 3, as $W$ seems to be used as both an element of $\mathbb{R}^{c_{out} \times c_{in} \times n \times n}$ and later as a square matrix. This is quite confusing. See also 7)

4) Page 3. Cite some standard reference where a formal statement of the convolution theorem is given.

5) Page 3. Mention that $F_n^\ast$ denotes the conjugate transpose.

6) Eq. 3. Explain how are $x$ and $X$ related

7) Similar to 3), $\operatorname{FFT} \colon \mathbb{R}^{n \times n} \to \mathbb{C}^{n \times n}$, but in Algorithm 1) this is applied to tensors. In this particular case one can hint what is going on, but again, it is unnecessarily difficult to track what lives what conventions are being used when an operator is applied to a tensor. This should be made explicit somewhere.

8) Page 4. "However, it can be extended to all orthogonal convolutions by multiplying by a diagonal matrix with $\pm 1$ entries (Gallier, 2006; Helfrich et al., 2018).".
This is not true. The correct theorem says that "for every $B \in \operatorname{O}(n)$ there exists a diagonal matrix $E$ with entries in $\{ -1, 1 \}$ such that $E\operatorname{cay}(X) = B$ for some $X \in \operatorname{Skew}(n)$. Note that the matrix $E$ depends on the matrix $B$. As such, to cover the whole $\operatorname{O}(n)$, (or $\operatorname{SO}(n)$) one would have to optimise over this discrete set, which is not easy.

9) Could the authors comment on how their method to parametrise non-square matrices via the Cayley transform relates to that used in the classical paper:

Zaiwen Wen and Wotao Yin. A feasible method for optimization with orthogonality constraints.Mathematical Programming, 142(1-2):397–434, 2013.

Note that the standard method would account for lifting the problem from $\operatorname{St}(n,k)$ to $\operatorname{SO}{n}$, compute the Cayley map and then just take the first $k$ columns. Their method is slightly different. Is your method one of these two or is it different to both of them? I believe that this discussion should be in the main paper in the related work section.

10) Just out of curiosity, how expensive is this method when compared with an unconstrained convolutional layer in wall-clock time?

***Experiments***

What is the reason for using Björck & Bowie (1971) to orthogonalise the layers in the other examples? This is a very bad way to do so which leads to some terrible dynamics. I believe that Cayley should be used throughout all the experiments. Could the authors report the results of the other methods while using Cayley also for them?

***Conclusion***

This is a very strong paper, with a very nice application of orthogonalisation in the context of CNNs. I think that the presentation of the content in the paper is very good. The only thing I did find quite confusing is the lack of definitions in section 4. I would be happy to increase my score provided that the problems raised here are addressed, in particular that of the experiments.

---

### Official Review · AnonReviewer3 · 2020-10-20
**Review: Important problem, recommend acceptance.**

**Rating:** 7
**Confidence:** 3

**Review:**

### Summary
**Objective:** attain orthogonal convolutions.

**Approach:** use $conv_W(x)-conv_{W^T}(x)$ which is skew-symmetric together with the Cayley map.

### Strengths
**[+]** Using transposed convolutions to attain a "skew-symmetric" convolution is interesting.

**[+]** The paper constructs an orthogonal convolution instead of enforcing orthogonality on a reshaped kernel. This ensures properties like gradient norm preservation. I always thought the lack of this property was a major limitation of previous work. The paper address this limitation for for circular convolutions.

### Weaknesses
**[-]** The resulting convolution is circular.

**[-]** They need invertibility which adds constraints on the convolutions stride and shape.

**[-]** The author could be more explicitly outline the above limitations wrt expressiveness.

### Recommendation: Acceptance 7
**[+]** The paper address an important problem, and present the first "real" orthogonal convolution.

**[-]** The orthogonal convolution is circular, and I think such limitations could be outlined more explicitly.

Before discussion: *I was torn between 6 and 7. If the authors add a paragraph that explain limitations wrt circular convolution, stride and shape I'll be happy to change to 7. That said, I'll condition this statement on first re-evaluating my opinion based on the comments from the other reviews. It is possible I am being a bit too harsh here, but it is also possible the other reviewers caught something I did not.*

The authors addressed my concerns and it seems my co-reviewers didn't bring new concerns to light, so I increased my score to 7.

### Questions and Concerns
**All my questions was addressed by the authors before I submitted my review**. These questions, and their answers, should be visible to all other reviewers, the area chairs and the program committee. For completeness, I copied the questions below, please note they were already answered by the authors.

---
Question 1. Equation (3) and (4) use both $x$ and $X$. Is this a typo, and should both be $X$?

Question 2. Equation (3) and (4) use $FFT:\mathbb{R}^{n\times n}\rightarrow \mathbb{C}^{n\times n}$ on $X\in \mathbb{R}^{c_{in}\times n \times n}$ and $W\in \mathbb{R}^{c_{out}\times c_{in} \times n \times n}$. Strictly speaking this is not well-defined, however, I suspect you just mean $FFT(X)[i, :, :]:=FFT(X[i, :, :])$. Is my suspicion correct?

Question 3. You write *"..., we see that the $(i, j)$'th Fourier domain output pixel is given by the matrix-vector product .. "*. This reminded me of the periodic convolution section 3.2 from [0]. Is the the same, if not, what is the difference?
I'm asking to make sure I understand your method. If it is the same, I believe your method can be summarized as done below. Please let me know if this is correct/wrong.

The periodic convolution $c(x)$ from [0] is invertible, which allows us to compute the inverse of the skew-symmetric operation $S(x)=c(x)-c^T(x)$. Since we can compute both $S(x)$ and its inverse, we can compute the Cayley transform of $S$ which is orthogonal.

I want to emphasize that, if this is what you're doing, I do think it is novel.

Question 4. To compute the inverse used in the Cayley transform you rewrite the convolution operation to matrix-vector multiplication in Fourier space. To do this you utilize the convolution theorem which only works for circular convolutions. How does this effect the resulting orthogonal convolution?
- (a) Does it have some circular behavior?
- (b) Must the kernel size be equal to the input size, e.g., W.shape=(c_in, c_out, **n**, **n**) with X.shape=(c_in, **n**, **n**)?
- (c) Must the convolution be unstrided?

Maybe I missed it in the paper, but such restrictions were not entirely clear to me.

---

### Additional Feedback
Fix the typo and maybe emphasize broadcasting notation in equations (3) and (4) as previously discussed.

---

### Official Review · AnonReviewer4 · 2020-10-25
**Another parameterization for Orthogonal Convolutional Layer**

**Rating:** 7
**Confidence:** 4

**Review:**

The paper provides another parameterization for orthogonal convolutional layers using the Cayley transform, different from BCOP. To the best of my knowledge, this parameterization is novel. However, I have a few questions regarding the proposed method.

(1) For 1D-convolutional layers, BCOP is a complete characterization. From the paper, the parameterization is not complete since the eigenvalues are all +1. While it is possible to multiply a diagonal matrix with either +1 or -1 entries, it is not clear such multiplication closes the gap. I am curious whether the composition is complete; otherwise, the proposed parameterization is strictly weaker than BCOP.

(2) For 2D-convolutional layers, I believe both BCOP and the proposed method using the Cayley transform are incomplete. So I am curious whether the proposed parameterization is a proper superset of BCOP. Without the argument, it is vague to state the proposed method is more expressive than BCOP --- the better results could come from optimization instead of parameterization. That being said, the paper is still interesting if the proposed parameterization covers a different subset of all orthogonal 2D-convolutional layers (i.e., neither a superset nor a subset of BCOP). In this case, the authors need to characterize the difference between these two subsets. Which layer can be parameterized by Cayley transform but not BCOP, and vice versa?

If the authors can clarify the questions, I will definitely increase my score. Others are minor comments:

(3) Comparing computational complexities for different methods, RKO, OSSN, SVCM, BCOP, and Cayley transform is desired.

(4) The BCOP includes the experiment of Wasserstein distance estimation. Empirically, it is better to show the proposed method is better than BCOP in various scenarios if a theoretical justification is too hard, if not impossible.

The questions above are well addressed in the response, and I would like to increase my score.

---

### Official Review · AnonReviewer1 · 2020-10-28
**A nice addition to the list of methods orthogonalizing convolutional layers but the method is not so clear to me**

**Rating:** 7
**Confidence:** 4

**Review:**

**Comments after rebuttal**: Overall the authors have addressed many concerns in the latest version of the paper. In the current form, I have a much better impression of this work and thus raise my score to a point where I can recommend acceptance. In the following I add some after-rebuttal comment at the end of each item of my original review:

## Summary:
This papers presents an application of the Cayley Transform to parameterizing a subset of orthogonal convolutional layers. As I understood, in contrast to other methods that orthogonalize the convolution by reshaping the kernel, the orthogonalization is performed in the frequency domain: the Cayley transform is applied to the kernel matrices corresponding to each pixel, after applying the FFT and then the output is recovered using the inverse FFT. Then it is shown empirically that this method preserves norms and that either this method or a combination of the Cayley transform and a method from a different paper (RKO) improves upon certified robustness (using the simple Lipschitz-margin bound). Some marginal gains on clean error are also reported.

## Pros:
**1. Quality: The empirical evaluation is good at comparing between baselines**, however it could be improved (see Cons).

**2. Originality: It seems interesting and novel to do orthogonalization using the Cayley Transform in Fourier domain, as this seems to bypass limitations of other methods that have troubles trying to "orthogonalize" the matrix corresponding to the convolution**. The main issue is that such process usually returns a matrix that does not correspond to an actual convolution and this paper seems to avoid that issue altogether.

**3. Significance: From experimental evaluation it seems that the proposed method is a strong candidate among other alternatives to orthogonal convolutional layers**. However it is also more expensive. **after-rebuttal:** the authors have added additional experiments and show that in practice their method can be as fast or faster than other baselines in certain scenarios.

**4. Significance: It seems that the space of orthogonal transformations parameterized in the way proposed in this paper can be more expressive than previous work BCOP or RKO**. Although this could be clarified further by the authors.

## Cons:
**1. Clarity: after reading the paper multiple times, It is not clear to me if this is a heuristic argument or if we can prove that the output of the so-called "Cayley Layer" is indeed orthogonal**. That is, can we show a theorem like "Let $f(x)$ be the output of the Cayley Layer (Algorithm 1), then $||f(x)\|/ ||x|| = 1$ for all $x$." Is this true? does it make sense? It looks like experiments in figure 1 are trying to validate this empirically but I wonder if this formal result could be obtained. **after-rebuttal:** The authors make this more clear in sections 3 and 4.

**2. Clarity: I feel that there are many terms thrown around without a clear definition, and it makes the arguments confusing.** In particular I am talking about the whole of section. I elaborate further:
- First, skew-symmetry is defined for square matrices and then the term "skew-symmetryze" is for the transformation $f(A)=A-A^T$. So what does skew-symmetryze mean? if it means a function such that its ouput is skew-symmetric then it would be trivial to output any constant skew-symmetric matrix. The important thing about $f(A)=A-A^T$ is that it is in fact the orthogonal projection of $A$ onto the space of skew-symmetric matrix (modulo a factor of 1/2).
- Then what does it mean to skew-symmetrize a convolution? here the authors use the notation $conv_{W}(x)$ which is not defined anywhere but I guess that it means the linear operator corresponding to the convolution with kernel $W$. but is this a single kernel matrix? I assume because otherwise what is $W^T$ in $\text{conv}_{W^T}$ note that in the previous paragraph $W$ is defined as a tensor of higher rank=4 so its transpose is not defined!
- In line with the previous, the vanilla way to "skew-symmetryze" $conv_W$ would be $conv_W - (conv_W)^T$. What is the relation with the expression $conv_W - conv_{W^T}$? In summary, I found this paragraph extremely confusing. **after-rebuttal:** The authors go over the definitions and now make more clear statements in section 4, which makes the argument more concise. The current version is more enlightening to the reader.

**3. Clarity: I am confused by the two proposed methods CRKO and Cayley which I think are not properly highlighted**. In the "Architecture considerations" in page 6 what do you mean that “for our Cayley layer… we use the Cayley transform for consistency” what is the alternative? isn’t using the CT the whole point about CRKO and Cayley Layer? what is the difference between the Cayley Layer and CRKO? On a first read I thought they were the same(?) Looks like CRKO is RKO + Cayley transform but It would be good to clarify the differences 

**4. Significance: It is not clear how the method deviates in practice from using the orthogonal layer mentioned in the work of Lezcano-Casado (more precisely in the repo https://github.com/Lezcano/expRNN) where it is mentioned that their Orthogonal layer can also be used with CNNs. Why isnt that approach compared here?** Is it possible to compare to it? It looks like the issue might be again that the Cayley-transformed matrix might not correspond to a convolution, could the authors confirm or comment on this? **after-rebuttal:** This has been further clarified by the authors, the method mentioned does not provide orthogonal convolutions.

**5. Significance. It is not clear how the methods and baselines improve over vanilla CNNs which should be included as a baseline, in my opinion, to understand the benefits of orthogonal layers (Table 1). Additionaly it is claimed that Cayley is state-of-the-art on clean test-accuracy and certified robustness. This is really debatable**, if clean test accuracy is the metric isn't adversarial training state of the art? I think it should also be included in Table 1. In any case the improvement over BCOP seems marginal. Additionaly, certified robustness is computed in a really limited way using the Lipschitz-margin bound whereas there are other methods that could certify better. So it is not clear what is the state-of-the-art claim and I believe this is over-stated. **after rebuttal**: I still believe the state-of-the-art claims are overstated because the authors do not compare to methods that certify robustness without resorting to Lipschitz continuity constraints. In my opinion, this should be downplayed and change to a more clear conclusion for example: among methods that certify robustness through the Lipschitz bound of CNNs using a per-layer regularization, our method achieves state-of-the art. this would be much closer to reality.

**6. Clarity: I did not understand the difference between some baselines**: Given that BCOP and RKO both use the Bjorck-Bowie algorithm would it be possible to explain the difference between those two methods? **after rebuttal**: clarified by the authors.

## Other comments
1. Page 5. I think the issue with OSSN is that dividing by the largest singular value DOES NOT correspond to the projection onto the 1-ball corresponding to the spectral norm. That is actually achieved by clipping the singular values as in SVCM. Of course this works only for matrices without additional constrainst, which is precisely the problem SVCM has with the additional constraint that the matrix should correspond to a convolution.

2. Final sentence in paragraph "Strided convolutions" (page 6): Typo: remove “used”  **after rebuttal:** fixed.

3. In line 6 of algorithm 1 I believe there is a typo. Should it be $\tilde{Y} - \tilde{A} * \tilde{Y}$ or am I missing something? **after rebuttal:** fixed.

4. Missing references: In related work there is a discussion about methods controlling the Lipschitz constant of networks "There have been other ideas for calculating and controlling the minimal Lipschitzness of neural networks..." there are some other interesting works in this area that can potentially lead to better certified robustness than using the vanilla upper bounds like in Tsuzuku et. al. (2018) see
https://openreview.net/forum?id=rJe4_xSFDB
https://arxiv.org/abs/2002.03657 **after rebuttal:** missing references have been added.

Overall, I think that this paper is valuable but some parts need to be improved to recommend acceptance. If some of these issues are adressed in a succint way then I would be willing to increase my score.

---

### Decision · Program_Chairs · 2021-01-07
**Final Decision**

**Decision:**

Accept (Spotlight)

**Comment:**

Very good paper: it proposes a novel parameterization of orthogonal convolutions that uses the Cayley transform in the Fourier domain. The paper discusses several aspects of the proposed parameterization, including limitations and computational considerations, and showcases it in the important application of adversarial robustness, achieving good results. The reviews are all very positive, so I'm happy to recommend acceptance.

Also, a big shout-out to the reviewers and to the authors for being *outstanding* during the discussion period. The reviewers engaged with the paper to a great depth, and the authors improved the paper considerably as a response. Well done to all of you.